# OrganoidTracker: Efficient cell tracking using machine learning and manual error correction

Rutger N. U. Kok[ID]¹, Laetitia Hebert[ID]², Guizela Huelsz-Prince¹, Yvonne J. Goos¹, Xuan Zheng¹, Katarzyna Bozek³, Greg J. Stephens[ID]²,⁴, Sander J. Tans¹,⁵*, Jeroen S. van Zon¹*

1 AMOLF, Amsterdam, The Netherlands, 2 Okinawa Institute of Science and Technology Graduate University (OIST), Onna-son, Okinawa, Japan, 3 Center for Molecular Medicine Cologne (CMMC), University of Cologne, Cologne, Germany, 4 Department of Physics and Astronomy, Vrije Universiteit Amsterdam, Amsterdam, The Netherlands, 5 Bionanoscience Department, Kavli Institute of Nanoscience Delft, Delft University of Technology, Delft, The Netherlands

* s.j.tans@tudelft.nl (SJT); j.v.zon@amolf.nl (JSV)

**Data Availability Statement:** Software is available at Github, https://github.com/jvzonlab/OrganoidTracker. The pre-trained neural network is available at the DANS repository, https://doi.org/10.

## Abstract

Time-lapse microscopy is routinely used to follow cells within organoids, allowing direct study of division and differentiation patterns. There is an increasing interest in cell tracking in organoids, which makes it possible to study their growth and homeostasis at the single-cell level. As tracking these cells by hand is prohibitively time consuming, automation using a computer program is required. Unfortunately, organoids have a high cell density and fast cell movement, which makes automated cell tracking difficult. In this work, a semi-automated cell tracker has been developed. To detect the nuclei, we use a machine learning approach based on a convolutional neural network. To form cell trajectories, we link detections at different time points together using a min-cost flow solver. The tracker raises warnings for situations with likely errors. Rapid changes in nucleus volume and position are reported for manual review, as well as cases where nuclei divide, appear and disappear. When the warning system is adjusted such that virtually error-free lineage trees can be obtained, still less than 2% of all detected nuclei positions are marked for manual analysis. This provides an enormous speed boost over manual cell tracking, while still providing tracking data of the same quality as manual tracking.

## Introduction

Increasingly, scientists are looking at development and homeostasis of organs at the single-cell level [1–3]. The invention of organoids, which are mini-organs grown in vitro that mimic the cellular structure and some of the functionality of an organ [4], has made it possible to study tissues at a level of detail that was not possible before. Nowadays, organoids are able to mimic an increasing number of organs with an improving accuracy and complexity [5, 6]. The ability to follow individual cells would allow one to directly reveal the historical genealogical relations

17026/dans-274-a78v. All other relevant data are within the paper and its Supporting Information files.

**Funding:** R.K. and X.Z were funded by an NWO Building Blocks of Life grant from the Dutch Research Council, number 737.016.009, https://www.nwo.nl/. G.H.P and J.S.v.Z were supported by an NWO Vidi grant from the Dutch Research Council, number 680-47-529, https://www.nwo.nl/. K.B., L.H. and G.J.S. were supported in the project by funds from OIST Graduate University, https://oist.jp/. Other authors received no specific funding for this work. The funders had no role in study design, data collection and analysis, decision to publish, or preparation of the manuscript.

**Competing interests:** The authors have declared that no competing interests exist.

between cells, their changing spatial organization, their cell divisions and deaths, and may be combined with markers for cell differentiation and other vital cellular processes. Elucidating these cellular spatio-temporal dynamics will be central to understanding organ development and tissue homeostasis, as well as associated pathologies.

Since the 1960's, researchers have used algorithms to automatically detect cells from microscopy images [7]. Traditionally, automated cell tracking algorithms have been rule-based, i.e. they relied on sets of discrete rules to detect cells. Such approaches have been successfully employed to automatically detect and track cells in highly complex, multicellular systems, such as early fruit fly, zebrafish and mouse embryos [8]. However, organoids often represent adult (epithelial) tissues and therefore pose specific challenges for automated cell tracking using rule-based approaches, compared to early embryogenesis. First, during cell divisions in adult epithelia, cell nuclei rapidly move from the basal to the apical side of the cell, and back again after the division [9]. This rapid nuclear movement is challenging as cell trackers typically rely on detection of fluorescently labeled cell nuclei for tracking. Second, in epithelia nuclei are closely-packed, with microscopy images often showing overlap in the fluorescence signal between adjacent nuclei due to limits in optical resolution. Finally, in adult epithelia cells rapidly turn over, which makes cell death a common occurrence [10]. The resulting cell debris has an visual appearance similar to nuclei of live cells, and hence must be ruled out as false positives. In recent years, systematic comparison of the performance of cell trackers has shown that convolutional neural networks outperform rule-based approaches in accuracy of cell detection [11]. Convolutional neural networks have already been used successfully for microscopy cell images in 2D [12–16] as well as in 3D [17–19]. Therefore, we decided to use convolutional neural networks as a basis for developing an approach to track most, if not all individual cells in time-lapse movies of growing organoids.

A key challenge for using convolutional neural networks is obtaining enough annotated microscopy data for training. Training data is typically generated by hand initially, and then often augmented by rotating, rescaling and varying the brightness and contrast of the images [18], or even by using neural networks that generate synthetic microscopy images to supplement the training data [19]. To detect fluorescently-labelled nuclei in time-lapse movies, two different approaches have emerged. In the first approach, neural networks are trained to segment individual nuclei [20]. This has the advantage that it provides information not only on cell position, but also nuclear shape and volume, that can be used to identify the same nucleus in subsequent frames. However, this comes with the disadvantage that creating manually segmented training data is very challenging. In the second approach, neural networks are trained to only predict the center locations of each nucleus [13, 17, 21]. While this provides little information on nuclear shape, generating training data is significantly easier, as only a single point needs to be annotated for each nucleus. Here, we follow the latter approach, allowing us to make use of an extensive collection of manually annotated imaging data generated by us. We combine it with the technique of providing the image input coordinates to the neural network [22], allowing for optimized cell detection for various depths within the image stack.

To link cell positions generated by the convolutional neural network into cell trajectories, and to correctly keep track of cell identities during cell divisions, we use a min-cost flow solver [23] that we further optimized for detection of cell divisions in organoids. Tracking all cells and their divisions in entire organoids raises the possibility of directly measuring cell lineages, e.g. the division patterns by which adult stem cells generate differentiated cells. To achieve this, cell detection and linking needs to take place with high accuracy, as a single switch in cell identities can have a profound impact on the resulting cell lineage. For that reason, we followed to approach in Refs. [24–26] and implemented tools for manual curation of the

automatic tracking data to eliminate such mistakes, with the software flagging tracking data that is likely erroneous for subsequent examination by the user.

We tested the ability of this approach to perform automated tracking of all cells in intestinal organoids, which are organoids that model the small intestinal epithelium [10]. The intestinal epithelium rapidly renews itself; in mammalians every five days the stem cells replace almost all of the differentiated cells. This results in a flow of cells from the stem cell compartment (crypt) to the compartment where fully differentiated cells die (villus). Intestinal organoids contain all cell types from the intestinal epithelium [27], making them a highly valuable model system for intestinal biology. Using our approach we could obtain tracking data of the same quality as the manually annotated training data, but for entire organoids and at vastly higher speed. Using the tracked data, we could for the first time establish complex cell lineages of up to five generations and visualize spatial patterns of cell division and cell death over the entire organoid. Such data will allow tissue-level study of cell divisions, movement and death at high time resolution, and will likely give new insight in how and when cells make their cell fate decisions.

## Results

### Overview of cell tracking approach

We are interested in tracking cells in time lapse videos of growing intestinal organoids. In the videos, we observe the nuclei of the cells which was made fluorescent using H2B-mCherry. The videos are up to 65 hours in length with the time resolution set to 12 minutes, resulting in videos of 84 to 325 time points. Each pixel is 0.32 x 0.32 μm in size, and the separation between two z-layers is 2 μm. As can be seen in Fig 1, the images have irregular fluorescence intensity, which is not only caused by intrinsic variations of nucleus fluorescence, but also by the location of the cells: some parts of the organoids are oversaturated, while parts deeper inside the organoids are undersaturated. Especially in the oversaturated parts of the images, the nuclei can appear to visually overlap. In addition, the organoids exhibit high nuclear density, with large variation in the shape, orientation and intensity of nuclei. Moreover, the requirement to limit phototoxicity results in low-contrast images. Finally, images suffer from low resolution in the z-direction. All of these factors make it challenging to automatically recognize nuclei.

The imaged area of the organoid typically contains over 200 cells, which makes manual tracking unfeasible. Instead, we had previously obtained (to be submitted) manual tracking data for a part of the imaged area, namely the crypt region. For that region, the center position of about 70 nuclei was recorded manually. The center position of each nucleus was gauged by eye and may therefore be a few pixels off. Additionally, links were created between the same nuclei in different time points, so that the nuclei could be tracked over time. In total, manual tracking was done for 2826 time points, coming from 17 different organoids.

In our cell tracking approach, we use a convolutional neural network for cell detection, a min-cost flow solver [23] for linking cells over time and a small set of rules to detect improbable cell tracking data that the user needs to verify and correct. This complete approach is displayed schematically in Fig 2. Step 1 is formed by the convolutional neural network: using raw images, it outputs images with peaks at the center location of the nuclei. This network needs to be trained using ground-truth data, which is done using the manual tracking data. Then, in Step 2 the cells are linked using a min-cost flow solver that can handle cell divisions [23], which optimizes for the smallest cell movement in between time points, for the smallest changes in nucleus volume in between time points and for the longest cell tracks. Finally, in Step 3 the program reports all cell tracks that it sees as suspicious based on a small set of rules,

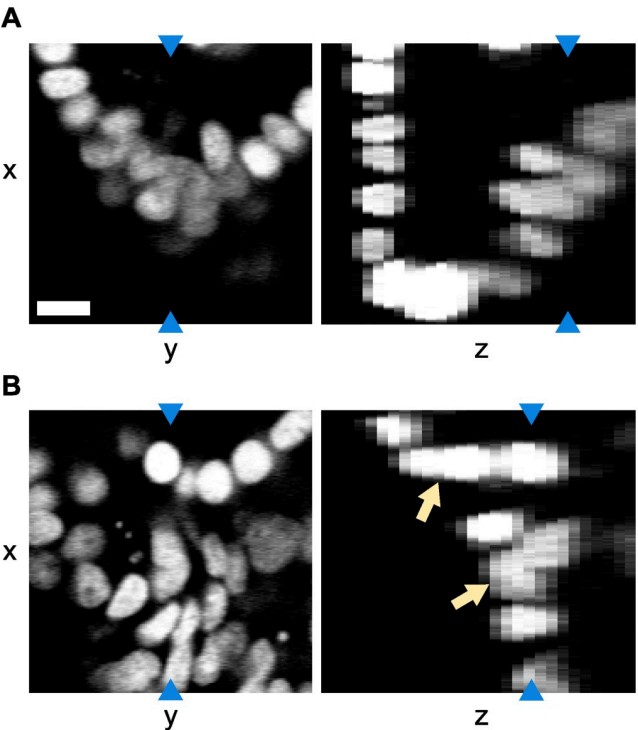

**Fig 1. Example of a tracked organoid.** Microscopy image slices in the xy and xz planes using H2B-mCherry to visualize the cell nuclei. Blue arrows indicate that both image slices in the same panel represent the same pixels. Note that the resolution in the z-direction is lower. The images show that there is both variation in intensity and visual overlap in the nuclei (indicated by the orange arrows), especially in the z-direction. The scale bar is 10 μm.

which the user needs to inspect. For us, this workflow resulted in cell tracks of the same quality as manual tracking though obtained vastly faster.

## Step 1: Automated cell detection

The nucleus center positions from our manually annotated tracking dataset were used to train a fully convolutional neural network, where the input and output are 3D images of the same shape. We trained the network with 3D microscopy images as the input images. Here, each image represents a single time point. The network architecture follows a standard U-Net [20] type architecture with all the convolutional layers replaced by the CoordConv layer [22] adapted for 3D images (Fig 3A). The CoordConv layer concatenates the absolute location of pixels to each convolution input. Compared to the simple convolution layer which is translation equivariant, this layer is useful for cases where the absolute position of the pixels is important to the task.

For the output images of the network, we want images where the brightness of each pixel corresponds to the chance of that pixel being a nucleus center. As multiple pixels close together can have a high likeliness of being a nucleus center, this results in mostly dark output images with bright spots at the locations of the nucleus centers. To make the neural network produce such output images given microscopy images at the input images, it needed to be trained on pairs of microscopy images and artificially generated output images. The generated output images are black images that contain bright, Gaussian-shaped spots with standard deviations of 2 px at the locations of the nuclei (Fig 3B). The locations of the nucleus centers were determined from the manual tracking data. The Gaussian shape of the spots acknowledges the fact

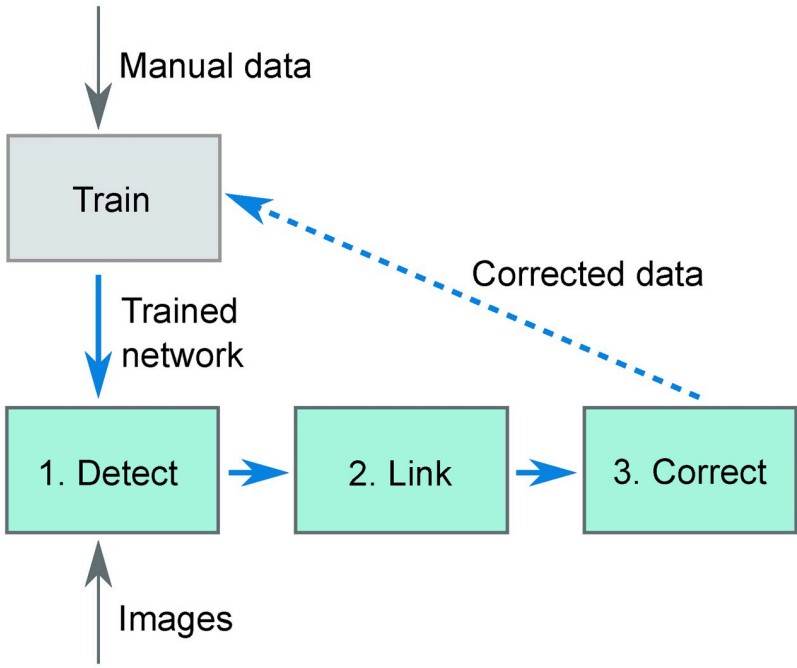

**Fig 2. Overview of the tracking software.** Using ground-truth data of nucleus locations in microscopy images (obtained from manual detection) a convolutional neural network is trained. This trained network is then used to detect (step 1) cells in new images. The detections are then linked over time (step 2), after which the user manually corrects the output with the help of an error checker (step 3). In principle, the manually corrected data can be used as additional ground-truth data to improve the training of the convolutional neural network, leading to an efficient, iterative improvement in performance of the convolutional neural network (dashed line).

that there is an uncertainty in the exact location of the nucleus center. Because the resolution in the z axis of the images is lower than in the x and y axes, the Gaussians for the labeled volumes are in 2D and don't cross several xy-planes.

We trained the network on 9 organoids, containing 1388 time points in total. For training, columns of 64x64 px from random locations within the images were used. This made the training algorithm use less memory compared to using the full 512x512 px, and also randomized the x and y coordinates of the nuclei, but unfortunately it does prevent the network from using information in the images that is located further away from the nucleus. To make the training more robust, random perturbations were applied to the training images [20]. The images were randomly flipped in the x, y and/or z direction, rotated a random number of degrees (0-360), scaled to 80%-120% of their original size, made brighter or darker with at most 10% of the maximum intensity in the image and finally the contrast was adjusted according to $I_{new} = (I - \langle I \rangle) \cdot c + \langle I \rangle$ where $I_{new}$ is the resulting intensity of a pixel, $I$ the original intensity of that pixel, $\langle I \rangle$ the mean intensity of all pixels in the image and $c$ the contrast factor, which varied from 0.5 to 1.5. We use a weighted mean squared error as the loss function between the network output and the labeled volume. Because the labeled volumes were mostly composed of zeroes, we gave more importance to the Gaussian spots by applying weights that correspond to the percentage of non zero values in the labeled volume.

Once the network was trained, it generated output images that show where the nucleus centers are located (Fig 3C). Each pixel in the 3D image represents the probability of that pixel being the nucleus center, resulting in a probability distribution with small peaks at the location of the nucleus centers. We interpolated linearly the empty space between the slices so that the

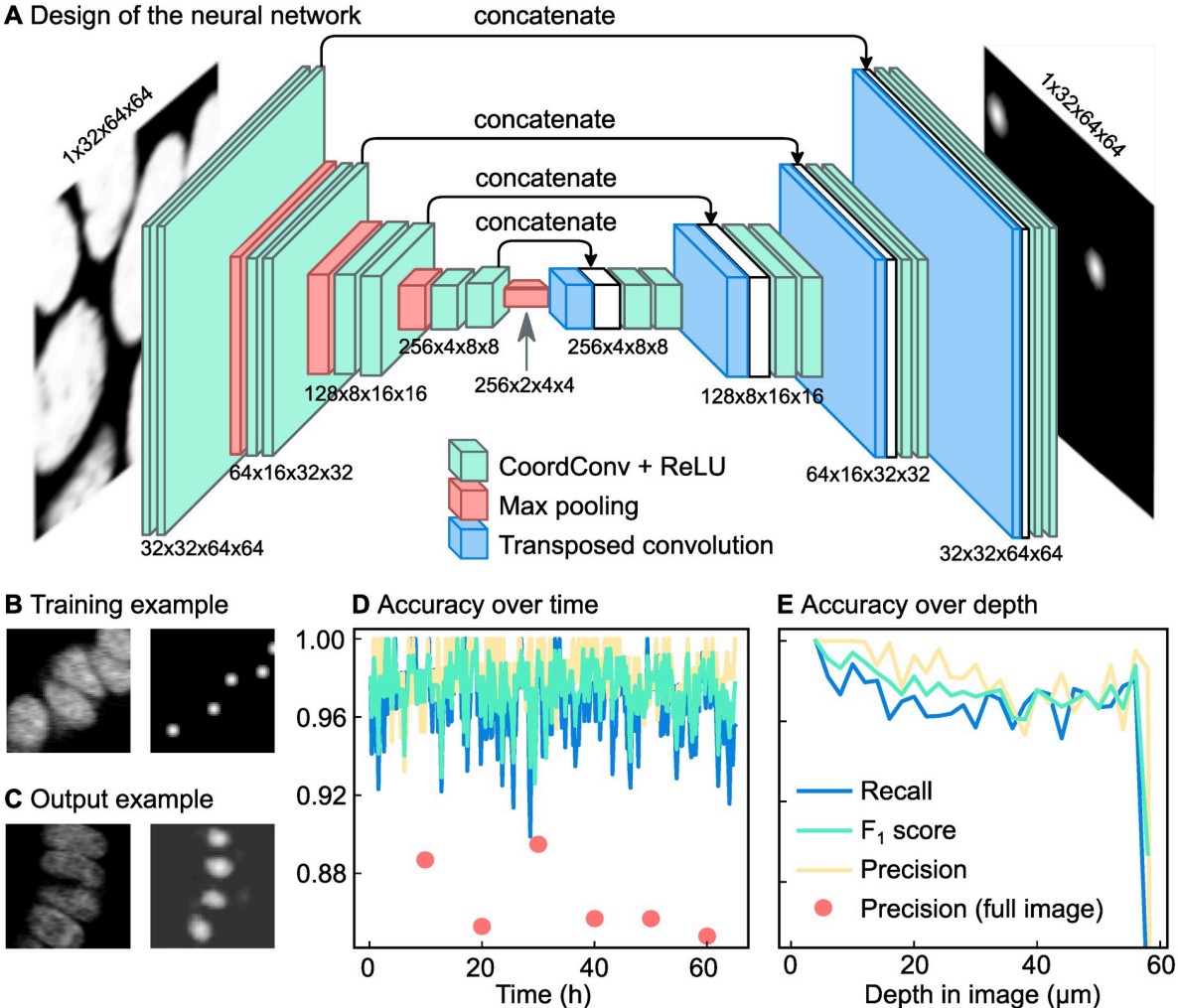

**Fig 3. Automated detection of cell nuclei by a convolutional neural network.** (**A**) Schematic overview of the network. The network is a standard convolutional network with absolute pixel coordinates added as input values. This makes it possible to adjust its detection network both for cells deep inside the organoid and cells near the objective. (**B**) Example of training data. On the left a slice of a 3D input image, on the right Gaussian predictions created from manual tracking data. The brightness of each pixel represents the likeliness of that pixel being the nucleus center. (**C**) Example of a network prediction, which shows the probability of each pixel being a nucleus center, according to the neural network. (**D**) Accuracy of the neural network over time compared to manual tracking data obtained in a subsection of a single organoid. For six time points (annotated with red circles) tracking data was compared to manually annotated data for an entire organoid. In this case, we observed a lower precision. For this organoid, the recall is 0.97 while the precision is 0.98. (**E**) Same data as in Panel d, now plotted over z. The accuracy of the network drops at the deepest imaged parts, likely because part of each nucleus falls outside the image stack here.

resulting volume had the same resolution in the z axis as in x and y. This allows us to apply a 3D peak detection algorithm (`peak_local_max` in scikit-image 1.1.0 [28]) to detect these local maxima in the interpolated 3D volumes. The resulting 3D coordinates are considered to be the locations of the nucleus centers in the full 3D volume. We then map back these coordinates to the closest image slice.

To evaluate the performance of the network, we needed to know how many of the detections are true positives or false positives, and how many false negatives there are. To do this, we compared the automatic tracking data to manual tracking data of 8 organoids (1438 time points) that were not used for training the neural network. Because these images are from

separate organoids, we can use this tracking data to evaluate the model generalization. One challenge in the performance evaluation was that it is difficult to measure the number of false positives from the neural network, as only 30% to 40% of all cells visible in the images were tracked. Therefore, at any location where the neural network reports the presence of a nucleus while the manual annotations do not, we cannot a priori be sure whether there is a false positive or whether that part of the image was simply not manually annotated. To overcome, we used the following approach.

Any nucleus center detected by the neural network was assigned to the closest nucleus center from the manually tracking data, under the condition that the distance was no longer than 5 μm. Every nucleus center cannot have more than one assignment. Each successful assignment was a true positive. Then, any manually tracked nucleus center that was left with no assignments became a false negative. Finally, any nucleus center from the neural network that was left with no assignments was regarded as a false positive if it was within 5 μm from a manually tracked nucleus center, otherwise it was rejected. This ensured that misdetections within the manually tracked area were still detected.

We measured three values to quantify the performance of the network: the precision, recall and the $F_1$ score. The precision is the fraction of the nucleus detections done by the neural network that were correct. Mathematically, it is defined as

$$\text{precision} = \frac{\text{TP}}{\text{TP} + \text{FP}} \tag{1}$$

with TP the number of true positives and FP the number of false positives. The recall is the fraction of nucleus detections in the ground truth that was detected by the neural network, and it is defined as

$$\text{recall} = \frac{\text{TP}}{\text{TP} + \text{FN}} \tag{2}$$

with FN the number of false negatives. For a neural network it is important that both the precision and the recall are high. In one extreme example, if a neural network would mark almost every pixel as a nucleus center, it would have a high recall, but a precision of almost zero. In the other extreme, if the neural network would only find back the most obvious nuclei, it would have a high precision but a low recall. To express the quality of a neural network as a single number, the harmonic average of the precision and recall was calculated, which is known as the $F_1$ score:

$$F_1 = 2 \cdot \frac{\text{precision} \cdot \text{recall}}{\text{precision} + \text{recall}} \tag{3}$$

The precision turned out to be 0.98 with a standard deviation of 0.007, which means that 98% of the detections done by the neural network were consistent with the manual data. The recall turned out to be on average 0.96 with a standard deviation of 0.02, which means that 96% of all manually tracked cells were detected by the neural network. Based on these values for the precision and the recall, the $F_1$ score of the network is 0.97.

Fig 3D displays the accuracy over time, which shows that the accuracy did not degrade over time, when the organoids have grown larger. As we can see in Fig 3E, the accuracy is the highest close to the objective and then drops before levelling off from 20 μm. In the deepest part of the organoid, the accuracy drops further. This is because the full nucleus is no longer visible; part of the nucleus now falls outside of the image stack.

The manually tracked region may not be representative for the whole organoid as there are also untracked regions which mostly consist of fluorescent material coming from dead cells. To check the performance of the network for the organoid as a whole, for six time points all nuclei were manually annotated in the entire image stack. This resulted in 1318 nuclei being annotated. The recall for these images was still 0.96, but the precisions, shown in Fig 3D as red dots, decreased to 0.87. By observation of the locations of false positives, we observed that this was mainly because dead cell material was recognized by the neural network as if it were a live cell nucleus. Together, this indicates that the recall is approximately consistent for the entire image, but that the precision is lower in regions with more dead cell material. To obtain correct cell tracking data, any spurious annotations of dead cell material will need to be filtered out later in the tracking process. This could be done either during the automatic linking step or during the manual error correction step.

In addition to calculating the $F_1$ score of the neural network for our organoids, we also calculated the detection accuracy (DET) score. The DET score is another measure for the accuracy of identifying cells [29]. The software available to calculate DET scores requires cell segmentation masks, not just cell positions as is the case for our cell tracker [11]. Therefore, we drew a pseudo-mask of the nuclei, which was a sphere with a of 5 μm around each detected position. In closely-packed regions of the organoid, this sphere is larger than the nucleus. Therefore, multiple spheres can overlap. If this happens, each pixel in the overlapping region of multiple spheres was assigned to the closest detected position. To make sure that detected positions outside the manually tracked area were not regarded as false positives, we removed all positions outside the manually tracked region. The approach was the same as for the $F_1$ score: any detected nucleus center more than 5, m away from a nucleus center in the manual tracking data was deleted. The DET score, calculated in this manner for the 8 organoids, was 0.93±0.03, with a score of 1.00 corresponding to error-free detection. These scores are in the high range of scores obtained by other trackers for developmental data sets of similar complexity [11]. Overall, these results indicate that our approach can accurately detect most cell nuclei in our organoids. The next step is to link the nucleus detections at different time points together, such that nuclei can be followed over time.

## Step 2: Linking of cell positions

To follow cells over time, the locations of the same cell nucleus at different time points need to be linked together. A link goes from a detected nucleus center at time point $t$ to the same nucleus center imaged at time point $t + 1$. Normally, every nucleus has one link to the next time point and one link to the previous time point. However, in case of a division a nucleus will split into two nuclei and therefore the nucleus will also have two links to the next time point. A straightforward way to create these links is to always assume that the nearest detected nucleus in the previous time point represents the same nucleus; this is called nearest neighbor linking. By going back in time, in theory we get detection of cell divisions for free: if two nuclei at time point $t + 1$ both have the same, single nucleus at time point $t$ as their closest nucleus, a division is generated.

Unfortunately, nearest-neighbor linking does not provide us with accurate lineage trees. We can see in Fig 4A that nearest neighbor linking creates unrealistically short cell cycles. Moreover, although rare, there is nothing that prevents a mother cell from having three or more daughters. Clearly, the assumption of the nearest nucleus always being the same nucleus does not hold. For this reason it is necessary to let the linking algorithm also consider links to nuclei further away than the nearest neighbor. In our method we consider all links that are at most two times further away in distance than the nearest neighbor. For most detected cell

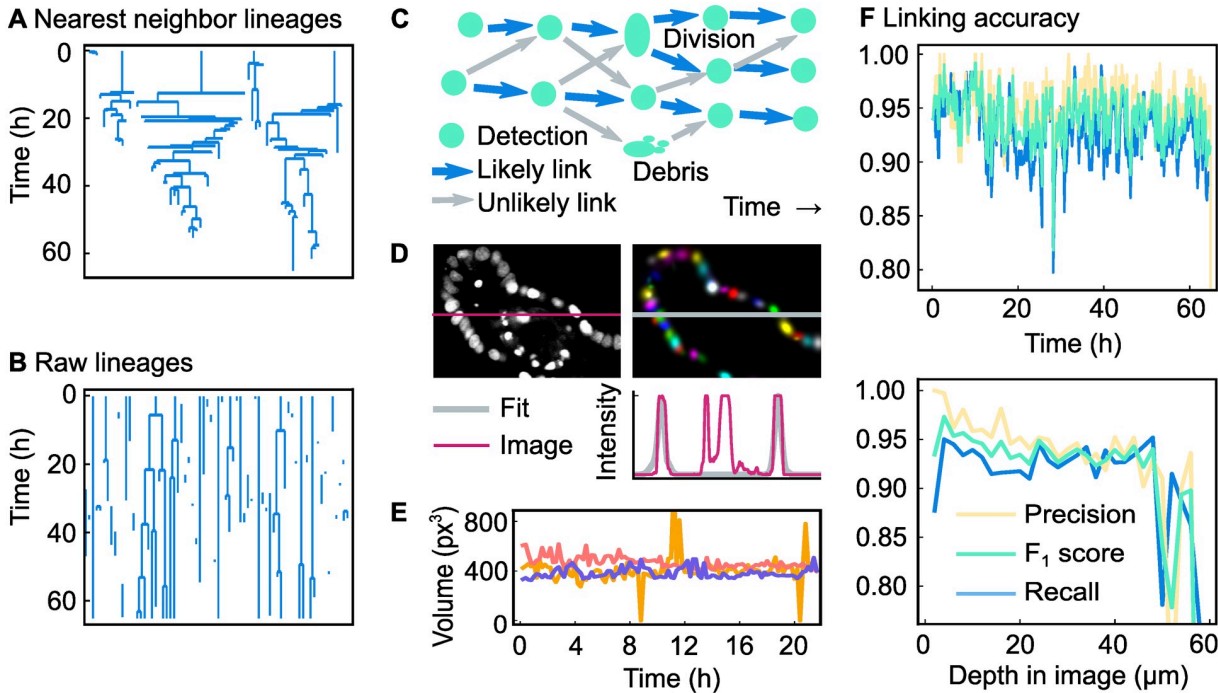

**Fig 4. Overview and results of the linking algorithm.** (**A**) Lineage trees obtained by linking using the nearest-neighbor method. (**B**) Raw lineage trees obtained by our linking algorithm. (**C**) Example of a network of links. There are many possibilities to link the cell detections in different time points together. Of all possible links (displayed as arrows), the most likely ones (displayed as blue arrows) are selected using a scoring system. (**D**) Original microscopy image, Gaussian fit and intensity profile of both. The intensity profile runs along the marked row in the image. We have chosen a section that highlights the fact that our approach can select nuclei and ignore cell debris, even though both have a similar fluorescence intensity profile. (**E**) Nucleus volume of three nuclei extracted from a Gaussian fit. The volume ($V = \mathrm{Cov}(x, x) \cdot \mathrm{Cov}(y, y) \cdot \mathrm{Cov}(z, z)$) is calculated for three non-dividing cells. The outliers are caused by errors in the Gaussian fit. (**F**) Accuracy of the linking algorithm presented here for a single organoid, over time and over the image depth. Note that the precision is higher than for nearest-neighbor linking.

positions, this results in multiple linking options for creating links (Fig 4C). Each of those options is given a score (an energy), of which the sum of all created links is minimized. This minimization is carried out using a min-cost flow solver adapted for tracking of dividing cells, developed by Haubold et al. [23]. Here, we used the following rules:

1. *An ordinary link is scored as score = $\Delta X + \sqrt[3]{\Delta V}$, with $\Delta X$ the change in position in μm, and $\Delta V$ the change in volume of the nucleus in μm$^3$* This ensures that links with the lowest change in position and volume are favored.

2. *Every track end is given a penalty of 1.* This favors creating longer tracks over multiple shorter tracks, which allows for the occasional long-distance link.

3. *Every track start is given a penalty of 1.* Again, this favors the creation of longer tracks.

4. *Every cell position that is ignored by the linking algorithm results in a penalty of 1.* This ensures that as much as possible of the data from the neural network is linked. If this rule would not exist, a globally optimal score could be reached by not creating any links at all.

5. *Cell divisions have a scoring system based on the change in volume and intensity.* This scoring system is described in S1 Appendix. A score below 1 means that a cell division here is unlikely, in which case no division is allowed here.

6. *If the cell division score is at least 1, the score from rule 1 is halved.* In the time around a cell divisions, the cell nuclei are expected to quickly move and change their volume, so it is appropriate to lower the penalty for position and volume changes.

The scores resulting from rules 1 to 5 are multiplied by a factor specified by the user of the software. Here, these factors are 20, 100, 150, 150 and 30, respectively. By changing these values, the scoring system can be tuned. For example, if the neural network accidentally detects dead fluorescent material as living cells, resulting in many short paths, this can be fixed by increasing the penalty for appearances and disappearances of cells. In this way, the high penalties at the beginning and end of the track together are larger than the penalty of ignoring those detected nucleus positions. As a result, the track is not created at all. To give another example, if the cells in a system tend to move slowly, the movement penalty can be increased, so that the likeliness of creating long-distance links is decreased. We used this in our system, to avoid creating a link to a neighbor nucleus when the correct nucleus was not detected by the neural network. For us this works well, because a cell track that abruptly ends will easily be found if the user rechecks all track ends, while a switch in cell identity resulting from a link to another cell is harder to detect.

For evaluating rule 1 and 5 it is necessary to know the volume of the nuclei. This volume is estimated using a 3D multivariate Gaussian fit. While nuclei do not always have a Gaussian-like shape, they can still be approximated by a Gaussian function in order to get an estimate of their volume [30]. Because we only want to fit Gaussian functions to living nuclei and not to fluorescent material coming from dead cells, we cannot fit the image as a whole. Instead, we need to separate dead cell material from the living nuclei, and only fit Gaussian functions to the latter. This is done by first performing a segmentation to divide the image into clusters, each containing zero, one or multiple cell positions detected by the neural network. The segmentation starts by performing an adaptive threshold on the original image $I_{image}$ to separate the foreground and background, resulting in a mask named $I_{mask}$. This mask is then improved by first removing all pixels from $I_{mask}$ that in the original image $I_{image}$ have a negative iso-intensity Gaussian curvature [30], which means that all pixels of lower intensity in between two places with higher intensity are removed. This is expected to remove the pixels in between two nuclei. Next, all holes in the mask are filled, resulting in the mask named $I_{mask,filled}$. On this mask, three rounds of erosion are performed with a 3x3x1 cross-shaped kernel on the mask. This step makes the area of foreground smaller, which makes the foreground fall apart into separate clusters. For that reason, this eroded mask was called $I_{mask,eroded}$. From the convolutional neural network, we know how many cells there are in each cluster and where their nucleus centers are. A Gaussian fit is then performed on $I_{mask,filled}$, which was the mask before erosion was applied. One Gaussian function is fit for each nucleus position and each Gaussian function is centered around the position given by the neural network, using a default intensity and a default covariance matrix. The following function is used to fit a single nucleus:

$$G(\vec{x}) = a\, e^{(\vec{x}-\vec{\mu})^T\, \Sigma^{-1}\, (\vec{x}-\vec{\mu})} \tag{4}$$

Here, $a$ is the maximum intensity of the function, $\mu$ is a vector representing the mean and $\Sigma$ is the covariance matrix. Together, these three variables describe the intensity, the position and the shape of the cell nucleus, respectively. $(\vec{x}-\vec{\mu})^T$ denotes that we are taking the transverse of $(\vec{x}-\vec{\mu})$, which results in a row vector. These functions are fitted to a blurred version of the original image, $I_{image,blurred}(\vec{x})$, so that the nuclei have a more Gaussian-like shape. For all pixels $\vec{x}$ in the cluster, the quadratic difference between that image and the outcome of the Gaussian

functions describing the nuclei is minimized:

$$\text{difference} = \sum_{\vec{x}} \left( I_{image,blurred}(\vec{x}) - \sum_i G_i(\vec{x}) \right)^2 \tag{5}$$

In Fig 4D, we can see an example of the Gaussian fit. The panel shows that not only the Gaussian fit correctly follows the intensity profile of the cells, but also that the fit ignores the debris of dead cells, which was previously filtered out by the neural network. To show the consistency of the Gaussian fit over time, we have plotted a volume ($\text{Cov}(x, x) \cdot \text{Cov}(y, y) \cdot \text{Cov}(z, z)$) over time for three arbitrary cells in Fig 4E. As can be seen in that figure, the Gaussian fit predicts a volume over time without significant noise. However, sometimes the fit predicts a volume more than twice the median volume, and sometimes only predicts a very low volume. This can happen if the fit of two nuclei results in one large Gaussian shape that covers both nuclei, and one very small Gaussian shape.

In Fig 4B the raw lineage trees obtained from the linking algorithm are displayed. By comparing both these lineage trees and lineage trees obtained from nearest-neighbor linking (Fig 4A) to lineage trees from manual tracking data (S1 Fig) it is clear that the method proposed here is more accurate than nearest-neighbor linking. To quantitatively measure the accuracy of our method, we need to calculate the precision (Eq (1)) and recall (Eq (2)). For this it is necessary to define true positives, false positives and false negatives in terms of links. We considered all detected nuclei at most 5 µm from the location of the manually annotated nucleus. If any of these nearby nuclei centers at time point $t$ had a link to any of the nearby nuclei centers at time point $t + 1$, this was considered to be the same link and hence a true positive. If no link existed between the nearby positions, this was considered to be a false negative. Naturally, every link in the manual tracking data can only be assigned to one corresponding link the automatic tracking data. Every remaining link in the automatic tracking data with no corresponding link in the manual tracking data was therefore in principle a false positive. However, as not the whole organoid was tracked, links were rejected if, at both time points, they did not have any nucleus centers in the manual tracking data within 5 µm.

Using this approach, we calculated the precision, recall and $F_1$ score for the same 8 organoids (1438 time points) as used for evaluating the detection performance. The recall of all organoids is 0.91 with a standard deviation of 0.04, the precision is 0.94 with a standard deviation of 0.02 and the $F_1$ score is therefore 0.93. Note here that the recall score is limited by the recall of the neural network: if the neural network fails to detect a nucleus, the two links of that nucleus will automatically also not be created. As can be seen in Fig 4F, the results are relatively constant over time, but not over depth: the accuracy is worse near the edge of the imaged area. In addition, we calculated the tracking accuracy (TRA) score, a different measure of how well cells are tracked in time [11, 29], using the approach outlined for calculation of the DET score. We obtained a TRA score of 0.92±0.04 averaged over 8 different organoids, with a score of 1.00 corresponding to error-free tracks. Like the DET score, these TRA score are in the high range of scores obtained by other trackers for developmental data sets of similar complexity [11]. Overall, our analysis shows that our automated cell detection and tracking approach yields cell tracking data of high quality, although not yet sufficient for error-free tracking. Hence, a subsequent step of manual data correction is required to achieve this.

## Step 3: Manual error corrections

While the lineages trees obtained from the previous step already show promising results, still the lineage trees are not correct: divisions are missed, cell are switched in identity, cell tracks

end prematurely and cell divisions are created where there are none. Based on these results, manual error correction is needed to obtain reliable tracking data. Time-wise, it would not be practical to revisit each nucleus at every time point. For this reason, the program flags all nuclei that violate one the following rules, so that the user can manually make corrections:

1. *Cells must have a link to the previous and to the next time point.* This detects cells that are appearing out of nowhere and cells disappearing into nothing. These cases usually indicate that a cell was not detected at a time point, or that a cell was detected where there was none. In case the cell actually dies, or if the cell goes out of the microscope view, the user can ignore this rule violation.

2. *At least 10 hours must have passed before a cell divides again.* A violation of this rule indicates that a cell identity switch has happened. The exact number of hours can be changed.

3. *Cells must not have moved more than 10 μm in between two time points.* Such a quick movement can point to a switch in cell identity. The exact number of micrometers can be changed.

4. *The nucleus volume of a dividing cell must be larger than the volume of the daughter cells combined.* This rule helps identifying misdetected divisions.

5. *The volume of a nucleus may not shrink more than 3 times.* This rule helps finding switches in cell identity and missed cell divisions.

6. *Two cells may not merge together, and cells may not have more than two daughters.* Even though linking does not allow these situations, this rule can be violated if a person makes a mistake while manually correcting the tracking data.

Any violations of the above rules are automatically flagged by an error checker runs as the last step of the linking process. The program guides the user through the list of errors, prompting the user to either suppress them, or to manually correct the tracking data. While correcting the tracking data, the error checker immediately checks every edit done by the user. As a result, if the user accidentally creates a cell division with three daughter cells, the user will immediately see a cross appearing on the mother cell. The strictness of the rules can be varied, with stricter rules leading to more false warnings, but a lower risk for missing tracking errors. When we adjusted the parameters such that we got tracking data of the same quality as manual tracking data, 1 to 2% of all cell detections needed to be manually verified.

To evaluate the efficiency and performance of the manual data correction step based on the above rules, we compared the output of our cell tracker before and after manual data correction against manually annotated data in our training data set for two organoids. Before data correction, the recall was 0.93±0.02 and the precision 0.95±0.01. After data correction, the first organoid produced identical lineage trees compared to the manual tracking data, while the other organoid contained one tracking mistake: the wrong cell was declared dead. To evaluate how many of the warnings were false positives, we quantified what percentage of these errors led to a change in the tracking data. If the annotated nucleus position was moved or deleted, or if there was a change in the locations of the nuclei it was linked to, the warning was considered to be a true positive. For both organoids, we found that 80% of the warnings were true positives.

## Comparison to existing methods

Currently, other methods exist that combine automated tracking with manual correction, albeit without using neural networks, with one example being TrackMate [25]. Using such a rule-based approach to cell detection, rather than using neural networks, has the advantage

that no large set of training data is required. As the creation of such datasets is often a time-consuming and laborious process, one could wonder if our effort to create such a training dataset was justified here. How well would TrackMate perform when applied on our images?

To test this, we used the so-called Downsampled Laplacian of Gaussian detector of Track-Mate, which is the only detector suitable for spots larger than 20 pixels, according to its documentation. The detector has three parameters, the spot size, the downsampling factor and the detection threshold. By manual optimization, we arrived at a spot size of 10 μm, a downsampling factor of 2 and a threshold of 1. Like for our neural network, we measured the performance by only considering the manually tracked region of the organoid. The recall was on average 0.66, with a standard deviation of 0.04 across the organoids. This means that in our hands, TrackMate missed a third of all cells in the image. For comparison, the neural network only missed 4% of all cells. The precision was higher: 0.969 with a standard deviation of 0.006, which is almost as high as the precision of the neural network (0.98). This indicates that Track-Mate is conservative at recognizing nuclei, but when it does recognize a nucleus, it is almost always correct. However, in total this results in an $F_1$ score of only 0.79, which is considerably lower than the score of the neural network, which was 0.97.

Even though one third of the nuclei were not detected, we still made an attempt at linking using the LAP tracker of TrackMate, using its default settings. For the 8 testing organoids together, the recall was 0.55 with a standard deviation of 0.04 and the precision was 0.79 with a standard deviation of 0.04. This results in an $F_1$ score of 0.65. This result means that almost half of the correct links are already missing. Extensive manual correction would be required to establish cell tracks at similar quality as we produced with our approach. Overall, these results underscore to importance of efficient and accurate cell detection, as we achieved here using neural networks.

## Analysis of tracking data

In Fig 5 we show the results of this whole-organoid tracking approach. First, this approach makes it possible to connect lineage dynamics to spatial position of cells in an organoid. In Fig 5A 2D slices of microscopy images of an organoid are displayed, with its 3D reconstruction shown in Fig 5B. In the reconstruction, the cells have been colored by lineage: selected cells were colored based on which cell they originated from in the first time point. Videos of both panels are included as S1 and S2 Videos. In Fig 5C lineage trees are shown of the same organoid and using the same coloring. We can see that cells of the same lineage tree tend to stay connected in space, although cells of multiple lineage trees do intermingle. In addition, we can see that the largest lineage trees reside at the lowest position in the crypt region, which is the growing protruded region in Fig 5B.

Next, we used our tracking data to study dynamics of cell divisions in space and time. We observed that almost all cell divisions are happening in the crypt (S3 Video), which is consistent with the literature [31]. We could see that at the bottom of the crypt, the number of divisions is the largest. However, there were a few cells at the bottom of the crypt that, in sharp contrast to their neighbors, did not divide, or divided only once. These are likely to be Paneth cells, which are differentiated cells that reside in the stem cell compartment [32]. In Fig 5D we saw that divisions were initially synchronized, while later in time this synchronization was less apparent.

## Software

Source code and software packages are available at https://github.com/jvzonlab/OrganoidTracker. Documentation including tutorials is available at the same address. The

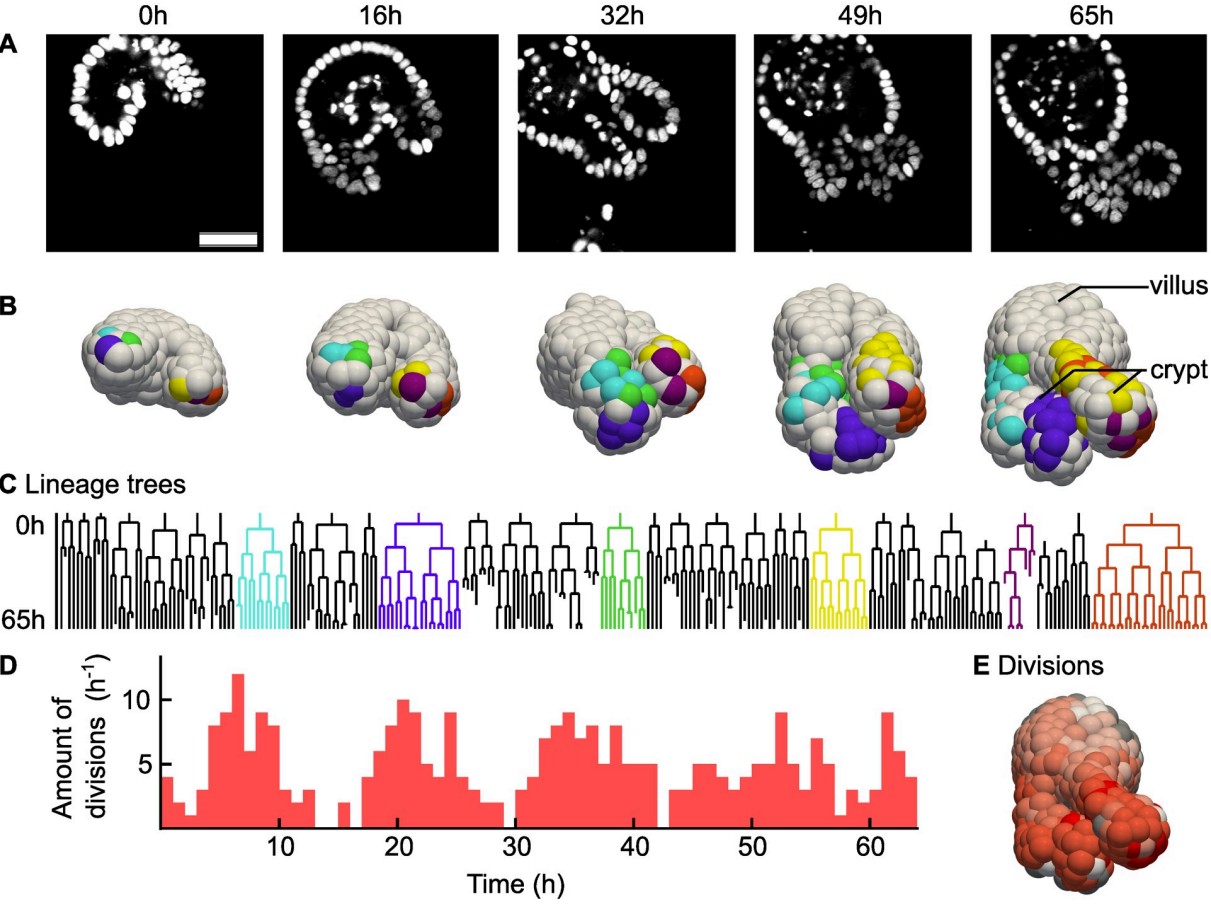

**Fig 5. Analysis of a fully tracked organoid.** (**A**) Single microscopy image slices of an intestinal organoid from a time lapse movie of 65 hours, in steps of 16.25 hours. H2B-mCherry was used to visualize the nuclei. Scale bar is 40 μm.(**B**) Digital reconstruction of the same organoid as in Panel a. Cells of the same color originate from the same cell at the beginning of the experiment. (**C**) Selected lineage trees. The colors match the colors in the reconstruction from Panel b. (**D**) Number of cellular divisions over time in a single organoid. (**E**) Number of times each cell divided during the time-lapse integrated over time, from 0 (white) to 5 (red). For gray cells, we could not count the amount of divisions because the cell was outside the image at an earlier point in time.

trained model used for this article has been made available in the DANS data repository: https://doi.org/10.17026/dans-274-a78v [33]. This model is suitable for 3D confocal images of cell nuclei, with a resolution of approximately 0.32 μm/px, z-separation of 2 μm and low background noise. Another model is available for trained for the lower resolution of 0.40 μm/px, still with the same z-separation and noise tolerance.

The released software is capable of manual tracking, training the neural network, using the neural network to detect nuclei in microscopy images, linking nucleus detections from different time points and assisting with the correction of automated tracking data. To correct the warnings, one would normally use an interface to jump from warning to warning. Instead, one can also jump from warning to warning within a single lineage tree. Additional tools are available to highlight all cells within a (sub)lineage, or to jump from a point in a lineage tree to the cell tracking data.

The software was written in Python 3.6, making use of the established software packages Mahotas 1.4.4 [34], matplotlib 2.2.2 [35], numpy 1.15.1 [36], OpenCV 3.4.1 [37], PyQT 5.9.2 [38] and Tensorflow 1.9.0 [39]. Image loading support is provided by tifffile 2020.2.16 [40] and nd2reader 3.1.0 [41]. The software saves all tracking data as JSON files.

3D visualization is possible by exporting the nucleus positions to Paraview [42], which can render those points as spheres. The software is also able to load and save files in the data format of the Cell Tracking Challenge [11], which makes it possible to exchange data with most other cell trackers. The software supports loading scripts written in Python, which can be used for data analysis. These scripts can be reloaded while the program is running, making it unnecessary to restart the program if you made a change to your analysis script.

## Discussion

We have developed a program for cell tracking based on nuclear markers inside organoids that uses a convolutional neural network for detection and a min-cost flow solver for linking cell positions. For our dataset 91% of all links in the ground truth were recognized. This is mainly because the neural network was only able to recognize 96% of the cells, and a single missing detection already means that the linking algorithm cannot reconstruct the cell track. However, the software facilitates manual error-correction so from this data cell lineages at the same quality as for manual tracking can still be obtained.

The convolutional neural network and the min-cost flow solver used for linking are both very generic: they make no assumptions on the microscopy pictures, other than that they are grayscale 3D images. In contrast, the Gaussian fit, which is used to provide an estimate of the nucleus volume and to recognize cell divisions, is quite specific to our images: it assumes bright nuclei on a dark background with not too much noise. Currently, eliminating the Gaussian fit is not possible, as the recognition of cell divisions depends on this.

As this approach uses a supervised form of machine learning, it is necessary to have a diverse set of training data available. Although the network will already automatically generate variations of your training data by adjusting the brightness, scale and rotation, still the network can have suboptimal performance for microscope images taken with different settings, like for example images with a different contrast or resolution. To solve this, one needs to retrain the network on training data created with those settings.

It is possible to think of further improvements to the algorithm. One could start training the neural network with a smaller training dataset, which results in a cell tracker of which its output requires a large amount of manual curation to get error-free results. After this correction step is over, it is then possible to use the manually corrected data to retrain the neural network. In this way, a feedback loop is created that improves the neural network every time more tracking data becomes available (Fig 2). In this paper no such feedback loop has been used: because we already had a large amount of training data beforehand, for us it was not helpful to retrain the network.

In general, we believe the way forward is to make the neural network as accurate as possible, and then give the linking algorithm the smallest possible amount of options to create links. As long as the biologically correct option is still part of this smaller set of options the linking algorithm can choose from, this approach would reduce the chance of the linking algorithm making mistakes. One possible step in this direction would be to let the neural network also classify cells: if the network could point out where the mother and daughter cells are, the linking algorithm could be restricted to only creating divisions at those locations.

In conclusion, we are now able to observe a growing organoid at the cellular level using cell tracking in microscopy videos. For this, we have developed a cell tracker using a convolutional neural network for cell detection, a min-cost flow solver [23] for linking of cell positions and a small set of rules to assist with data curation. It is our hope that cell tracking, with its coupling of lineage tree dynamics to time and space, will lead to in a better understanding of organoid growth and maintenance.

## Supporting information

**S1 Fig. Full lineages of the manual tracking data of the same organoid displayed in Fig 5.**
Red crosses denote the positions where a cell death was observed in the microscopy images.
This figure will also become part of an upcoming publication.
(PDF)

**S1 Video. Microscopy video of a growing organoid.** Pixels are colored by depth. Stills of this
organoid were included in the main text (Fig 5A).
(MP4)

**S2 Video. Reconstruction of an organoid using spheres.** Manually selected lineages have
been colored. Stills of this video were included in the main text as Fig 5B.
(MP4)

**S3 Video. Visualization of dividing cells in an organoid.** Nuclei are again drawn as spheres,
but are now colored by the time to the next division. The closer the next division, the redder
the sphere of a nucleus becomes.
(MP4)

**S1 Appendix. Cell division scoring system.** Explanation and equations of the scoring system
used to determine whether a given nucleus is a mother cell.
(PDF)

## Author Contributions

**Conceptualization:** Rutger N. U. Kok, Laetitia Hebert, Katarzyna Bozek, Greg J. Stephens, Jeroen S. van Zon.

**Data curation:** Rutger N. U. Kok, Guizela Huelsz-Prince, Yvonne J. Goos, Xuan Zheng.

**Formal analysis:** Rutger N. U. Kok, Laetitia Hebert.

**Funding acquisition:** Greg J. Stephens, Sander J. Tans.

**Investigation:** Guizela Huelsz-Prince, Yvonne J. Goos, Xuan Zheng.

**Methodology:** Rutger N. U. Kok, Laetitia Hebert, Guizela Huelsz-Prince.

**Project administration:** Jeroen S. van Zon.

**Resources:** Guizela Huelsz-Prince, Yvonne J. Goos, Xuan Zheng.

**Software:** Rutger N. U. Kok, Laetitia Hebert, Katarzyna Bozek.

**Supervision:** Greg J. Stephens, Sander J. Tans, Jeroen S. van Zon.

**Validation:** Laetitia Hebert.

**Visualization:** Rutger N. U. Kok, Laetitia Hebert.

**Writing – review & editing:** Rutger N. U. Kok, Jeroen S. van Zon.

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
