## [Decision Letter · Decision Letter 0]

23 Jun 2020

Pécs, Hungary

June 22, 2020

PONE-D-20-14769

OrganoidTracker: efficient cell tracking using machine learning and manual error correction

PLOS ONE

Dear Dr. Kok,

Thank you for submitting your manuscript to PLOS ONE. After careful consideration, we feel that it has merit but does not fully meet PLOS ONE’s publication criteria as it currently stands. Therefore, we invite you to submit a revised version of the manuscript that addresses the points raised by the Reviewers, listed below.

We look forward to receiving your revised manuscript.

Kind regards,

Joseph Najbauer, Ph.D.

Academic Editor

PLOS ONE

Journal Requirements:

Reviewers' comments:

Reviewer's Responses to Questions

**Comments to the Author**

1. Is the manuscript technically sound, and do the data support the conclusions?

Reviewer #1: Yes

Reviewer #2: Yes

2. Has the statistical analysis been performed appropriately and rigorously? 

Reviewer #1: Yes

Reviewer #2: Yes

3. Have the authors made all data underlying the findings in their manuscript fully available?

Reviewer #1: Yes

Reviewer #2: Yes

4. Is the manuscript presented in an intelligible fashion and written in standard English?

Reviewer #1: Yes

Reviewer #2: Yes

5. Review Comments to the Author

Reviewer #1: Kok et al. describe a method based on convolutional neural networks (CNNs) for the segmentation of cell nuclei, applying it to 3D organoid data. The problem is in general a challenging one, and the authors' tools will likely be useful. The paper is very clearly written, in terms of both text and figures. The methods used are not particularly novel, but this is not a criterion for PLOS ONE, which I fully support. Moreover, the methods are quite recent in their origin, so having additional examples of their use in the literature is valuable.

The one major flaw of the paper, however, is that it does not adequately survey past work, or explain similarities and differences between this approach and other approaches. It is very hard for the reader to get a sense of whether they should adopt these methods without a comparison.

More specifically:

- The authors note that cell trackers exist [2, 8, 9], but this only gets one sentence. Ref 8 compares several methods: how do these differ from the authors'? Ref 8 also provides 'test data,' I believe -- how do the authors' methods perform when applied to this data?

- The authors dismiss Ref. 10 (Amat 2014) with "For example, for tracking cells in fruit flies, zebrafishes and mouse embryos there are existing software packages" -- I fail to see how the nature of the organism matters. Is there something different about the *images* that makes these tools fail for organoid images? The images in Amat et al's paper are, in fact, 3D images of cell nuclei -- the same thing being imaged here.

- Some of the papers [11-17] seem to be about similar sorts of images to the authors' . What is different about either the images, the algorithm performance on similar data, or other features?

- There are many recent papers that are likely relevant, but not discussed; the authors should describe several of these, and compare with at least a few if these also involve similar microscopies and cell types. There are for example OrgaQuant: Human Intestinal Organoid Localization and Quantification Using Deep Convolutional Neural Networks (Kassis et al 2019, https://www.nature.com/articles/s41598-019-48874-y), Nucleus segmentation across imaging experiments: the 2018 Data Science Bowl (Caicedo 2019, https://www.nature.com/articles/s41592-019-0612-7.pdf?draft=collection), Volumetric Segmentation of Cell Cycle Markers in Confocal Image (Khan 2019, https://www.biorxiv.org/content/10.1101/707257v1.full.pdf); DeepSynth: Three-dimensional nuclear segmentation of biological images using neural networks trained with synthetic data (Dunn et al. 2019)

Reviewer #2: This is a very nice straightforward paper. Rutger et al. proposes here a method

to detect, "segment" and track nuclei in time-lapse videos displaying growth of

intestinal organoids. The detection is based on convolution network that is

trained to produce small bright spots near nuclei centres, nuclei are then

fitted with Gaussian to estimate their volumes (precise boundary is not extracted

nor required). Then, published solver is used to establish the tracking links

as a solution to a set of straightforward rules. Finally, a set of rules is

considered to alert an user of potentially spurious tracking (the rules

violating) loci and allow to curate it.

The method is described nicely, and I think I could reproduce it, besides it is

publicly available on Github. The paper reads very well, the problem is clearly

given, well motivated, well explained and evaluated. I could only find a few

typos (please, see below).

I have, unfortunately, a major concern about scientific novely of the paper.

Pardon me to say: it "only" describes a success story -- an application of

carefully tuned pipeline of otherwise known steps applied on one type of data.

While I appreciate (and see very well) the non-trivial amount of work beneath

it all, the nice software and the manuals to it!, what new lesson can we learn

after reading it? There is, for instance, no comparison against any other

state-of-the-art method. Or, since manual curation is here an acknowledged part

of the method, I was missing an evaluation of how efficient are the

quality-checking rules after the tracking? Does it emit a lot of false positives

(requests to check what is already good), does it have false negatives (does not

report what should have been reported), how much does the result improve after

the curation? Could the method/SW be compared against, e.g., TrackMate or Mamut

that both also do some cell detection, tracking and offer a GUI curation?

I don't want to kill the paper. I'm therefore trying to propose relevant topics

that could, in my opinion, enrich the study:

Regarding the evaluation of the detection and tracking, and since the authors

claim the availability of the export module to the Cell Tracking Challenge

data format, could the, to some extent standard methods, of DET and TRA be

also used to evaluate the tracking? This could illustrate the difficulty of the

problem in the context of other cell tracking tasks.

I think the strength of the paper is not necessarily the tracking itself, I

think it is more the environment around it, the SW: its ease of use, flexibility,

how to train the detector for another data, how to obtain some initial

annotation (can the proposed SW do it at all?), can one re-track only a subset

of the data, or just one lineage tree (starting from one cell, the daughter

cells may end up around each other or in some symmetry pattern; and anything

different is likely a reason for the inspection of the tracking?), how easily

can one manipulate the created lineage etc.

There was, I believe, a passage about re-training the network after first round

of detections, does it help at all?

I have tried some months back also to implement a rules to describe what is a

good track and what would allow me to point at problematic tracks. I've tried to

look for sudden change of trajectory direction, sudden change in displacement

distance of a cell between consecutive time points -- none of that worked well

(lot of false positives) for my embryonic developmental data. The best test,

still not ideal, for me was to detect how many different n-nearest neighbors are

there between consecutive time points. A comparison of similar rules, or

proposal of some good options, would be indeed a novelty... I am not aware of a

literature about precisely this. Could it be formulated for neural networks and

have them to detect anomalies?

What I also found an interesting aspect of the work is the proposed solution how to

evaluate performance of a method on only a partially annotated real data.

Immediately a question arises: To what extent does the choice of the annotation

subset influences the obtained performance level. In other words, how much does

the obtained performance represent a performance when measured against the full

annotation. The authors have 6 frames with full annotation, perhaps that could

be used a basis for such an experiment.

Minor, typps:

Fig 2. "corrected data _that_ can be",

L(ine)77 "ground-truth data, which _is_ done using",

L128 "apply a a 3D"

L193 spacing between words

Fig 4D, an intensity profile of the data and of the fit would be worthwhile to

appreciate the accuracy of the fit

6. PLOS authors have the option to publish the peer review history of their article (what does this mean?). If published, this will include your full peer review and any attached files.

Reviewer #1: No

Reviewer #2: No

---

## [Author Response · Author response to Decision Letter 0]

14 Aug 2020

Note: we have also attached our response in a separate file, which might be easier to read.

Reviewer #1: 

> Kok et al. describe a method based on convolutional neural networks (CNNs) for the segmentation of cell nuclei, applying it to 3D organoid data. The problem is in general a challenging one, and the authors' tools will likely be useful. The paper is very clearly written, in terms of both text and figures. The methods used are not particularly novel, but this is not a criterion for PLOS ONE, which I fully support. Moreover, the methods are quite recent in their origin, so having additional examples of their use in the literature is valuable.

> The one major flaw of the paper, however, is that it does not adequately survey past work, or explain similarities and differences between this approach and other approaches. It is very hard for the reader to get a sense of whether they should adopt these methods without a comparison.

> More specifically:

> - The authors note that cell trackers exist [2, 8, 9], but this only gets one sentence. Ref 8 compares several methods: how do these differ from the authors'?

We agree with the Reviewer that we could have better described the current cell tracking approaches in the field and how our method differs from and build on these results. In response to this comment, and a similar comment by Reviewer #2, we have extensively rewritten and expanded this section of the introduction, see lines 24-42 of the manuscript with track changes enabled. 

Briefly, Ref. [8] presents the results of three editions of the Cell Tracking Challenge, featuring 21 competing cell tracking algorithms, both classical approaches (employing sets of rules to detect and track cells) and machine learning approaches. The overall outcome was that methods based on machine learning, including convolutional neural networks, showed the best performance.

OrganoidTracker is similar to two methods in Ref. [8] (FR-Ro-GE and HD-Hau-GE). It uses a convolutional neural network similar to FR-Ro-Ge. One notable difference is that OrganoidTracker detects just the center positions of the nuclei, instead of the entire nucleus segmentation, facilitating the use of manually annotated data to train the network. In addition, we have added CoordConv layers to allow more optimized detections for nuclei at different z positions. For tracking, OrganoidTracker uses a probability graph-based optimization, similar to HD-Hau-GE. However, our method allows for efficiently correcting tracking data by hand, which is not available for the methods in Ref. [8], which as we show is crucial to establishing correct cell lineages.

> Ref 8 also provides 'test data,' I believe -- how do the authors' methods perform when applied to this data?

The Cell Tracking Challenge provides a range of datasets, both 2D and 3D, and both sparsely tracked and fully tracked. To address the reviewer’s question, we have downloaded the fully-annotated 3D dataset that featured the largest number of cells, which is the Fluo-N3DH-CE dataset containing time lapses of developing C. elegans embryos. As for all test data sets in Ref. [8], only two time-lapse movies with ground truth were available.

We trained our neural network on one data set and then tested it on the second. This resulted in a low precision of cell identification (recall of 0.98, precision of 0.40 and F1 score of 0.57), mostly due to a large number of false positives at the early embryo stage when the number of nuclei is small. At later time points the precision reached values >0.8. This likely reflects the limited number of data sets and annotated nuclei in the Fluo-N3DH-CE dataset compared to the manually annotated organoid data we used. For this reason, we have decided not to include these results in our manuscript.

> - The authors dismiss Ref. 10 (Amat 2014) with "For example, for tracking cells in fruit flies, zebrafishes and mouse embryos there are existing software packages" -- I fail to see how the nature of the organism matters. Is there something different about the *images* that makes these tools fail for organoid images? The images in Amat et al's paper are, in fact, 3D images of cell nuclei -- the same thing being imaged here.

We regret our poor wording here: it was not our intention to dismiss Ref. [10]. We agree that we did not specify clearly enough what the specific challenges of intestinal organoids are compared to these other systems. We have edited the manuscript to address this at lines 18-21 in the manuscript with tracked changes.

Briefly, while the data in Ref. [10] shares many challenges with our organoid data (crowded nuclei, uneven signal intensity, and a low signal-to-noise ratio), adult epithelia exhibit the following unique challenges:

1. Fast nuclear movement. During cell divisions, nuclei rapidly move from the basal to the apical side of the epithelium. As a result, there is often no overlap in the mask of the nuclei between two subsequent frames. Especially if multiple nuclei divide close together, as is the case in the stem cell compartment of intestinal organoids, this poses a complex challenge for automated tracking.

2. Closely packed cell nuclei. In epithelia nuclei are closely-packed with often visual overlap between adjacent nuclei due to due to limited optical resolution. At least by our visual inspection, crowding in intestinal organoids is worse compared to the embryos in Ref. [10].

3. Cell death. Adult epithelia continuously shed cells at the end of their life cycle. This results in cellular debris that is similar to nuclei in appearance, and hence must be ruled out as false positives (See for example Fig. 4d).

> - Some of the papers [11-17] seem to be about similar sorts of images to the authors' . What is different about either the images, the algorithm performance on similar data, or other features?

We have addressed this more clearly the rewritten and expanded section in lines 24-34 of the manuscript. 

Briefly, the networks in Refs. [11-14] are built for 2D images. These introduced the core principles on which most fully convolutional networks are built, but cannot directly be used for our 3D data. In contrast, Refs. [15-17] all use 3D U-net based Convolutional Neural Networks (CNNs) that are very similar to what we use, each with their own added functions. Ref. [15] focuses mostly on synthesizing extra training data for segmenting nuclei. Ref. [16] instead uses only nuclei center annotations, which makes it easier to create training data. Ref. [17] uses data augmentation (rotation, resizing, mirroring and rescaling) to generate extra training data. In our manuscript, we combine the approaches of Ref. [16] and [17]. We go beyond Ref. [16] and [17], which only focus on cell detection, by the addition of automated cell tracking and manual curation of tracking data.

> - There are many recent papers that are likely relevant, but not discussed; the authors should describe several of these, and compare with at least a few if these also involve similar microscopies and cell types. There are for example OrgaQuant: Human Intestinal Organoid Localization and Quantification Using Deep Convolutional Neural Networks (Kassis et al 2019, https://www.nature.com/articles/s41598-019-48874-y), Nucleus segmentation across imaging experiments: the 2018 Data Science Bowl (Caicedo 2019, https://www.nature.com/articles/s41592-019-0612-7.pdf?draft=collection), Volumetric Segmentation of Cell Cycle Markers in Confocal Image (Khan 2019, https://www.biorxiv.org/content/10.1101/707257v1.full.pdf); DeepSynth: Three-dimensional nuclear segmentation of biological images using neural networks trained with synthetic data (Dunn et al. 2019)

We thank the reviewer for providing us with these references. We have added these to our rewritten literature section on convolutional neural networks in lines 34-61 in the manuscript file with tracked changes enabled.

OrgaQuant is created to quantify the size of entire organoids in brightfield microscopy images. Although it shows that there is interest in observation of organoids, the software is not relevant for single-cell tracking. For that reason, we have decided to not include it in our manuscript.

The Data Science Bowl 2018 article shows an important example of U-net neural network in action for various 2D microscopy data sets. The Khan 2019 article uses U-net neural network for tracking on 3D cell data, similar to our work here, but focusing on plants where the density of nuclei is significantly lower compared to our organoid data. They do share our approach of tracking nuclei centers, rather than segmenting full nuclei, to facility creating of training data. We have added these references to our introduction.

The DeepSynth shows how to synthesize 3D microscopy images and use those as additional training material. We previously cited an earlier conference paper about DeepSynth, Ref. [15] in the previous version of the manuscript, but have now replaced this citation with the reference provided by the reviewer.

Reviewer #2:

> This is a very nice straightforward paper. Rutger et al. proposes here a method to detect, "segment" and track nuclei in time-lapse videos displaying growth of intestinal organoids. The detection is based on convolution network that is trained to produce small bright spots near nuclei centres, nuclei are then fitted with Gaussian to estimate their volumes (precise boundary is not extracted nor required). Then, published solver is used to establish the tracking links as a solution to a set of straightforward rules. Finally, a set of rules is considered to alert an user of potentially spurious tracking (the rules violating) loci and allow to curate it.

> The method is described nicely, and I think I could reproduce it, besides it is publicly available on Github. The paper reads very well, the problem is clearly given, well motivated, well explained and evaluated. I could only find a few typos (please, see below).

> I have, unfortunately, a major concern about scientific novely of the paper. Pardon me to say: it "only" describes a success story -- an application of carefully tuned pipeline of otherwise known steps applied on one type of data. While I appreciate (and see very well) the non-trivial amount of work beneath it all, the nice software and the manuals to it!, what new lesson can we learn after reading it? There is, for instance, no comparison against any other state-of-the-art method.

The main purpose of our manuscript is to present a method that allows tracking of all cells and their lineages in organoids. This had so far not been achieved, partly due to the unique challenges of adult epithelia we now outline in lines 24-34 of the manuscript with track changes, and is likely of interest to the large community working on organoids.

However, we appreciate the points that the reviewer makes here. Regarding scientific novelty, we have now more clearly described how our approach differs from and builds on other cell tracking approaches (lines 43-72). In addition, we have now provided more information on the performance of the manual correction and the quality of the resulting cell tracks and lineages (see points further below for details). While comparing performance against other methods is in general challenging, because getting optimal results from any cell trackers requires a significant amount of trial and error and manual labor, we have attempted to make such comparisons, as also detailed below. We hope that these revisions adequately address the reviewer’s concerns. 

> Or, since manual curation is here an acknowledged part of the method, I was missing an evaluation of how efficient are the quality-checking rules after the tracking? Does it emit a lot of false positives (requests to check what is already good), does it have false negatives (does not report what should have been reported), how much does the result improve after the curation?

Indeed, the referee is correct that the manual curation is a key part of our method. We have included more detail on the three questions the referee raised. Specifically:

1) False positives. We tested the warning system for two organoids. We considered a warning a false positive if no change was made to the tracking data for either the location of the nucleus, the identity of the cells it was linked to, or the location of the nuclei it was linked to. We found that for both organoids ~80% of warnings led to corrections in the tracking data, meaning that ~20% of the warnings were false positives. Hence, the rate of false positive warnings is already quite low. Note however, that evaluating a warning requires little user time and hence even a relatively large rate of false positives is not an issue.

2) False negatives. We tested how much the lineage trees of the automatically tracked and corrected organoid match the lineage trees of the manually tracked crypt of the same organoid. For both organoids, the lineage trees match manual tracks, with one exception in total, where a cell death was assigned to the wrong cell.

3) Improvement after curation. For both organoids, the recall and precision were between 0.9-0.95, both before correction. As mentioned in point 2), after curation the data was of practically the same quality as manual tracking.

We have added this information to the manuscript, see lines 444 – 455 in the document with tracked changes.

> Could the method/SW be compared against, e.g., TrackMate or Mamut that both also do some cell detection, tracking and offer a GUI curation?

We agree with the Reviewer that TrackMate, Mamut and our OrganoidTracker all offer complete tracking solutions, with automatic detection, automatic linking and manual curation included.

In terms of workflow, OrganoidTracker is the most similar to TrackMate. Both have detection, linking and curation steps. TrackMate supports both 2D and 3D tracking and offers to the ability to automatically filter tracks that might be erroneous, e.g. because cells move too rapidly. However, it does not offer an interface to correct all lineages, going through the warnings one by one, which is necessary for us to track all the cells. MaMut is mainly a tool for manual cell tracking and track visualization, and not well-suited for automated cell tracking. (In fact, the MaMut website specifically states that their automated tracking approach “must be considered as a strongly suboptimal tracking method, whose goal is strictly to assist manual annotation”). Therefore, we only attempted to compare the performance of TrackMate in tracking our organoid data. 

However, in our hands, we could only get TrackMate to recognize 70% of our nuclei, compared with manual data of a single organoid. To obtain this result, we used the Downsample Laplacian of Gaussian detector, which according to their documentation is the only detector suitable for spots larger than 20 pixels. We think this again underscores the power and flexibility of convolutional neural networks in cell detection. Because the TrackMate results appeared unpromising, we did not further explore this and decided to not include these first results in the revised manuscript.

> I don't want to kill the paper. I'm therefore trying to propose relevant topics that could, in my opinion, enrich the study:

> Regarding the evaluation of the detection and tracking, and since the authors claim the availability of the export module to the Cell Tracking Challenge data format, could the, to some extent standard methods, of DET and TRA be also used to evaluate the tracking? This could illustrate the difficulty of the problem in the context of other cell tracking tasks.

The DET score assesses the detection performance of our cell tracker, while the TRA score assesses the performance of linking the nuclei over time. We calculated both scores using the software at http://celltrackingchallenge.net/evaluation-methodology/ . The software requires two datasets: the ground truth, for which we used manual tracking data, and the output of the cell tracker.

To calculate the score, the cell tracker needs to generate a segmentation mask for every nucleus. Because our cell tracker cannot do this, we have simply drawn spheres with a radius of 5 µm. Where the spheres overlap, each pixel is assigned to the nearest nucleus center. In addition, we deleted every detected nucleus center more than 5 µm away from any nucleus center in the ground truth. This was done so that cells detected outside the manually tracked region are not regarded as false positives.

For the 8 testing organoids, the DET score was 0.93 ± 0.03 (average ± st.dev.) and the TRA score was 0.91 ± 0.03. These scores are in the high range of scores obtained by other trackers for similar developmental data sets. At the same time, as we explained above, these high DET and TRA scores are not sufficient for error free tracking and lineage analysis and further manual data correction is required to achieve this. Please see line 260-274 of the file with tracked changes for the additions for DET and line 307-409 for TRA.

> I think the strength of the paper is not necessarily the tracking itself, I think it is more the environment around it, the SW: its ease of use, flexibility, how to train the detector for another data, how to obtain some initial annotation (can the proposed SW do it at all?),

We agree very much with this assessment of the referee. Currently, the approach we present here has become the workhorse in our lab for analysis of organoid data. Consequently, it is indeed optimized for ease of use and flexibility. Indeed, we use this exact approach to train our algorithm on new data: for different types of organoids, different reporter lines in organoids and data taken on different microscopes. The ability to manually track cells, which is indeed part of our software, is crucial to generate the training data for this. We have described this aspect now more clearly in lines 484-491.

> can one re-track only a subset of the data, or just one lineage tree (starting from one cell, the daughter cells may end up around each other or in some symmetry pattern; and anything different is likely a reason for the inspection of the tracking?),

The software cannot retrack only a subset of the tracking data. The optimization procedure, as it is written now, optimizes all links to minimize the energy function. This reflects the fact that the position and dynamics of neighboring cells is often required (even when doing manual analysis) to identify a nucleus in subsequent frames. Specifically, daughter nuclei are not arranged spatially in a way that unambiguously links them to their mother: within epithelia cell divisions often result in substantial spatial rearrangements of nuclei.

> how easily can one manipulate the created lineage etc.

In the GUI, we optimized manipulation and correction of cell lineage data to facility rapid data correction. Overall, one can manipulate each individual link between nuclei in subsequent frames. To efficiently check warnings, the interface allows jumping from warning to warning. One can instead also jump from warning to warning within a single lineage tree. Additional tools are available to highlight all cells within a (sub)lineage, or to jump from a point in a lineage tree to the cell tracking data. We have modified the article at lines 484-491 in the manuscript with tracked changes to address this.

> There was, I believe, a passage about re-training the network after first round of detections, does it help at all?

While we suggested this potential improvement in the Discussion (lines 526-534), retraining did not help for the manually annotated dataset that we used in this manuscript, likely because the initial amount of data was already sufficient by itself. However, for dataset that we have generated since then using other microscopes, or with different organoids systems, we are using this approach successfully.

We did realize that our presentation in Fig. 2 might give the incorrect impression that retraining after manual data correction was an integral part of our approach. We have change Fig. 2 and its caption to clarify this.

> I have tried some months back also to implement a rules to describe what is a good track and what would allow me to point at problematic tracks. I've tried to look for sudden change of trajectory direction, sudden change in displacement distance of a cell between consecutive time points -- none of that worked well (lot of false positives) for my embryonic developmental data. The best test, still not ideal, for me was to detect how many different n-nearest neighbors are there between consecutive time points. A comparison of similar rules, or proposal of some good options, would be indeed a novelty... I am not aware of a literature about precisely this. Could it be formulated for neural networks and have them to detect anomalies?

In our hands, the graph-based optimization method for links already resulted in few cell identity switches. This method minimizes changes in distance in combination with other constraints. The most significant challenges for cell linking concern cell divisions, cell deaths and failed cell detections. In our hands, such events give rise the majority of warnings reported by the cell error checker for manual confirmation.

Currently, neither the linking system nor the error checking system uses a neural network. While we have thought about using our cell linking data (as opposed to just the nuclear position data) to train neural networks to track cells, this is a challenging problem. We are now working on improving cell linking by training classifier neural networks to predict the likelihood that a cell will undergo a division, or has recently divided, based on the appearance of a cell nucleus (e.g. rounding up of nucleus, breaking up in chromosomes). The ability to identify likely mother or daughter cells by appearance would allow us to constrain the cell linking data with much higher fidelity and thereby prevent the most frequent occurrence of cell identity switches for our data. 

> What I also found an interesting aspect of the work is the proposed solution how to evaluate performance of a method on only a partially annotated real data. Immediately a question arises: To what extent does the choice of the annotation subset influences the obtained performance level. In other words, how much does the obtained performance represent a performance when measured against the full annotation. The authors have 6 frames with full annotation, perhaps that could be used a basis for such an experiment.

This is an interesting suggestion by the reviewer. We agree that the measured cell detection performance could be different when comparing the result from the tracking software against manually annotated data from different parts of the organoid. For instance, the crypt and the villus show different challenges: in the crypt there are cell divisions that need to be recognized correctly, while the villus is in much closer proximity to cell debris that yield false positives in cell detection.

For that reason, we attempted to compare the automatic tracking data against a rectangular region in the villus section of the organoid, that consisted of about 50 cells. The performance in the epithelium was good: the precision was 0.98. However, we noticed that there were false positives in the lumen that were ignored for in the precision score, as we calculate it, because they were detected more than 5 micrometers away from any data in the ground truth. If we instead do not ignore any positions, but simply compare the annotations within the rectangle, the recall becomes unrealistically low: 0.93, leading to a precision of 0.89. This is because cell centers that were manually annotated just inside the rectangle, but were (correctly) detected just outside, are now registered as false negatives. 

We feel that including these results would only confuse the reader, as they are difficult to interpret correctly. From observing the locations of false positives and false negatives, we can already conclude that the number of false negatives is about the same everywhere in the epithelium, with little difference between crypt or villus regions, but that false positives are more prevalent in the lumen. We have made this more clear in the article: see line 252-259 in the file with tracked changes.

> Minor, typps: 

> Fig 2. "corrected data _that_ can be",

> L(ine)77 "ground-truth data, which _is_ done using",

> L128 "apply a a 3D"

> L193 spacing between words

We thank the reviewer for spotting these typos, and have corrected them. 

> Fig 4D, an intensity profile of the data and of the fit would be worthwhile to appreciate the accuracy of the fit

Fig 4D has been modified to add the requested intensity profile. We have chosen a section that highlights the fact that our approach can efficiently select nuclei and ignore cell debris, even though both have a similar fluorescence intensity profile. The text has also been changed at line 364-367 of the file with tracked changes to describe this new profile.

---

## [Decision Letter · Decision Letter 1]

11 Sep 2020

Pécs, Hungary

September 11, 2020

PONE-D-20-14769R1

OrganoidTracker: efficient cell tracking using machine learning and manual error correction

PLOS ONE

Dear Dr. Kok,

Thank you for submitting your manuscript to PLOS ONE. After careful consideration, we feel that it has merit but does not fully meet PLOS ONE’s publication criteria as it currently stands. Therefore, we invite you to submit a revised version of the manuscript that addresses the points raised by Reviewer #2, listed below.

We look forward to receiving your revised manuscript.

Kind regards,

Joseph Najbauer, Ph.D.

Academic Editor

PLOS ONE

Reviewers' comments:

Reviewer's Responses to Questions

**Comments to the Author**

1. If the authors have adequately addressed your comments raised in a previous round of review and you feel that this manuscript is now acceptable for publication, you may indicate that here to bypass the “Comments to the Author” section, enter your conflict of interest statement in the “Confidential to Editor” section, and submit your "Accept" recommendation.

Reviewer #1: All comments have been addressed

Reviewer #2: All comments have been addressed

2. Is the manuscript technically sound, and do the data support the conclusions?

Reviewer #1: Yes

Reviewer #2: Yes

3. Has the statistical analysis been performed appropriately and rigorously? 

Reviewer #1: Yes

Reviewer #2: Yes

4. Have the authors made all data underlying the findings in their manuscript fully available?

Reviewer #1: Yes

Reviewer #2: Yes

5. Is the manuscript presented in an intelligible fashion and written in standard English?

Reviewer #1: Yes

Reviewer #2: Yes

6. Review Comments to the Author

Reviewer #1: The authors have done a good job of clarifying the relationship between this work and past work, as well as describing the particular challenges of organoid images. I wish the discussion in the text had the level of detail of the "response to reviewers," though I realize that this would perhaps have been distracting. I support the publication of the paper.

Reviewer #2: The authors have addressed all questions, issues and suggestions of this

reviewer in their revised manuscript (and I thank you for that). The

manuscript is now even more clear. Well done. The changed parts of the

manuscript align well in the manuscript.

If I were to suggest final set of changes, I would ask for the following minor

improvements or clarifications:

Regarding the requested comparison to the TrackMate and Mamut, I thank the

authors for doing this extra work and for sharing with me the new findings.

I would like to encourage them to actually include this piece of

information into the manuscript. Not only it shows another "plus points"

for the SW of the authors, I think it is indeed a fair (despite "only"

quantitative) comparison of SW that include the same stages (detection,

linking _and_ curation) -- and that's useful piece of information also for

the community.

Regarding the conversion of nuclei detection (blobs) into

pseudo-segmentation masks (by placing spheres of 5 um radius), why were

some detections (and seg. masks) removed? It biases the measurement, no?

Please clarify in the manuscript if the filtering was due to the fact the only six

time points are fully annotated (in which case the filtering makes sense).

What is the average diameter of nuclei, is it less than 10 um? Otherwise,

the spheres are over-segmenting and bias the DET score.

In Fig 2, perhaps the first parentheses on line 2 could be "(obtained from

manual detection)"... detection is less difficult activity than tracking,

while it is enough for creating training data for the network... just an idea.

Line 8 (of the document with marked changes): "the the historical"

Line 48: "in in time-lapse movies."

PS: I apologize to the first author for mixing up his first name and surname

(in my first review I wrongly wrote "Rutger et al.").

7. PLOS authors have the option to publish the peer review history of their article (what does this mean?). If published, this will include your full peer review and any attached files.

Reviewer #1: No

Reviewer #2: No

---

## [Author Response · Author response to Decision Letter 1]

2 Oct 2020

Reviewer #1:

> The authors have done a good job of clarifying the relationship between this

> work and past work, as well as describing the particular challenges of organoid

> images. I wish the discussion in the text had the level of detail of the "response

> to reviewers," though I realize that this would perhaps have been distracting. I

> support the publication of the paper.

We thank the reviewer again for taking the time to review our paper, and we are happy that the reviewer supports the publication of the manuscript.

Reviewer #2:

> The authors have addressed all questions, issues and suggestions of this

> reviewer in their revised manuscript (and I thank you for that). The

> manuscript is now even more clear. Well done. The changed parts of the

> manuscript align well in the manuscript.

> If I were to suggest final set of changes, I would ask for the following minor

> improvements or clarifications:

> Regarding the requested comparison to the TrackMate and Mamut, I thank the

> authors for doing this extra work and for sharing with me the new findings.

> I would like to encourage them to actually include this piece of

> information into the manuscript. Not only it shows another "plus points"

> for the SW of the authors, I think it is indeed a fair (despite "only"

> quantitative) comparison of SW that include the same stages (detection,

> linking _and_ curation) -- and that's useful piece of information also for

> the community.

We have followed the suggestions of the reviewer and added a section ‘Comparison with other methods’ to the manuscript (lines 421-452 in the file with tracked changes), describing our cell detection and linking results for TrackMate. Briefly, for cell detection TrackMate displayed a low recall for our test data set of 8 organoids – on average, one third of all cells in an image are not detected. This relatively poor result makes linking challenging, which we confirmed when we attempted linking using the LAP tracker of TrackMate with default settings. 

> Regarding the conversion of nuclei detection (blobs) into

> pseudo-segmentation masks (by placing spheres of 5 um radius), why were

> some detections (and seg. masks) removed? It biases the measurement, no?

> Please clarify in the manuscript if the filtering was due to the fact the only six

> time points are fully annotated (in which case the filtering makes sense).

> What is the average diameter of nuclei, is it less than 10 um? Otherwise,

> the spheres are over-segmenting and bias the DET score.

We indeed filtered only because six time points are fully annotated. The text has been edited to clarify this, see lines 223-235 in the manuscript with tracked changes.

In less crowded areas we think a sphere of 10 µm is, at least on average, an accurate representation of a nucleus. However, in crowded regions the nuclei are often ellipsoidal. In those regions, the nuclei are arranged in a closely-packed epithelium, with the long dimension larger than 10 μm and the short dimension smaller. This results in a risk of oversegmentation, resulting from overlapping spheres. We prevented oversegmentation by assigning each pixel in the overlapping region of two spheres to the nearest nucleus center. As a result, the masks no longer end up as spherical, but actually start to look like the actual mask of the nucleus.

> In Fig 2, perhaps the first parentheses on line 2 could be "(obtained from

> manual detection)"... detection is less difficult activity than tracking,

> while it is enough for creating training data for the network... just an idea.

We thank the reviewer for this suggestion, and have implemented it.

> Line 8 (of the document with marked changes): "the the historical"

> Line 48: "in in time-lapse movies."

We thank the reviewer for spotting these errors, and have fixed them.

> PS: I apologize to the first author for mixing up his first name and surname

> (in my first review I wrongly wrote "Rutger et al.").

---

## [Editor Report · Decision Letter 2]

5 Oct 2020

Pécs, Hungary

October 5, 2020

OrganoidTracker: efficient cell tracking using machine learning and manual error correction

PONE-D-20-14769R2

Dear Dr. Kok,

We’re pleased to inform you that your manuscript (R2 version) has been judged scientifically suitable for publication and will be formally accepted for publication once it meets all outstanding technical requirements.

Kind regards,

Joseph Najbauer, Ph.D.

Academic Editor

PLOS ONE

---

## [Editor Report · Acceptance letter]

13 Oct 2020

PONE-D-20-14769R2 

OrganoidTracker: efficient cell tracking using machine learning and manual error correction 

Dear Dr. Kok:

I'm pleased to inform you that your manuscript has been deemed suitable for publication in PLOS ONE. Congratulations! Your manuscript is now with our production department. 

Kind regards, 

on behalf of

Dr. Joseph Najbauer 

Academic Editor

PLOS ONE